# A Special Ancient Bronze Sword and Its Possible Manufacturing Technique from Materials Science Analysis

**DOI:** 10.3390/ma15072491

**Published:** 2022-03-28

**Authors:** Chi Xie, Chunlin Fu, Sishi Li, Lingmin Liao, Guantao Chen, Chunxu Pan

**Affiliations:** 1MOE Key Laboratory of Artificial Micro- and Nanostructures, School of Physics and Technology, Wuhan University, Wuhan 430072, China; xiechi083@whu.edu.cn (C.X.); qianxun.li@whu.edu.cn (S.L.); 2School of Art, Royal Melbourne Institute of Technology University, Melbourne, VIC 3001, Australia; s3699080@student.rmit.edu.au; 3Materials and Structural Department, Changjiang River Scientific Research Institute, Wuhan 430010, China; stone142@163.com; 4Center for Archaeometry, Archaeological Institute for Yangtze Civilization, Wuhan University, Wuhan 430072, China; guantaokg@yahoo.com.cn; 5Center for Electron Microscopy, Wuhan University, Wuhan 430072, China

**Keywords:** composite bronze sword, copper-tin alloy, element diffusion, Fick’s law, ancient bronzes

## Abstract

In this study, it was found that an ancient bronze sword had special microstructures, i.e., a tin (Sn)-rich layer (Sn: 38.51 wt.%), that was around 0.1–0.3 mm in thickness in the bronze substrate (Sn: 18.57 wt.%). This sword was unearthed from the same *Chu* tombs of the “Sword of Gou Jian”, and dated back to the late Spring and Autumn Period (496 BC–464 BC). The experimental and theoretical analyses revealed that (1) the Sn-rich layer exhibited higher microhardness (around 650 HV) than the sword body (around 300 HV); (2) the Sn-rich layer showed a brittle fracture due to the formation of a large amount of α + δ eutectoid, while the sword body was of good toughness due to a large amount of α-Cu solid solution phase; and (3) theoretical calculations of Sn diffusion in the Cu substrate indicated that this Sn-rich layer could have been formed within several hours or several days if the temperature was above 600 °C. Therefore, this sword was proposed to be a novel kind of composite bronze sword, and the possible manufacturing technique was a surface treatment called “dip or wipe tinning” or tin amalgam, which was widely used in the Bronze Age. Technically, this process possesses more advantages than the well-known two-times casting for making the “double-colour” or bi-metallic composite bronze sword. This research showed that the materials processing level was beyond our expectations for ancient China 2500 years ago.

## 1. Introduction

The ancient Chinese people worshipped the bronze and iron swords, where they reached a point of magic and myth, regarding the swords as “ancient holy items”. Because they were easy to carry, elegant to wear and quick to use, bronze swords were considered a status symbol and an honour for kings, emperors, scholars, chivalrous experts, merchants, as well as common people during ancient dynasties. For example, Confucius claimed himself to be a knight, not a scholar, and carried a sword when he went out. The most famous ancient bronze sword is called the “Sword of Gou Jian”. It was unearthed from the No. 1 *Chu* tomb at Wangshan village in Jiangling city of Hubei province, China, in 1965. The sword body was decorated with diamond-shaped dark stripes, and there are eight concave Chinese Niao-Zhuan characters that means “King Gou Jian of Yue, Self-used Sword” with gilding in two lines. This sword was dated back to the late Spring and Autumn Period (496 BC–464 BC).

However, due to its simple shape and size, the process of making a bronze sword was simple when compared with other ancient bronze wares, let alone the “Chimes of Marquis Yi of Zeng” and the very complicated “Plate (*Zunpan*) of Marquis Yi of Zeng”. In other words, the “Sword of Gou Jian” became famous mainly because it was a “Sword of the King”, as well as idioms and legends of King Gou Jian’s “stoop to conquer”.

As is already known, ancient bronze weapons were cast from a Cu-Sn alloy. In general, a high Sn content corresponds to high hardness, but it is brittle and, thus, easy to break, while a low Sn content corresponds to high toughness, but it is soft and not sharp. In the Bronze Age and before the steel weaponry appeared, making bronze weaponry with both hardness and toughness was a big challenge.

Studies and archeological excavations indicated that the “double-colour” or bi-metallic composite bronze sword was a significant innovation of the two vassal states, namely, *Wu* and *Yue,* during the Spring and Autumn and the Warring States Periods [1,2,3,4,5]. The principle was to adopt a kind of two-times casting, i.e., first using bronze with a low Sn content (red and yellow colour) to cast the middle part of the sword ridge to obtain toughness and flexibility, and then using bronze with a high Sn content (yellow and white color) to mould the blade for high hardness and strength, as shown in Figure 1. Consequently, a bronze sword with good performance regarding both hardness and toughness was obtained.

In our previous work [6], two bi-metallic bronze *Ges* (dagger-axes), which belonged to a halberd manufactured during the Warring States Period (475 BC–221 BC), were discovered and studied. However, different from the composite bronze sword, the ridges of the *Ges* were made of pure Cu.

In this study, a special kind of ancient bronze sword was found that had a tin (Sn)-rich layer that was around 0.1–0.3 mm in thickness on the surface of the bronze substrate, and the Sn-rich layer had much higher hardness than the sword body. Regarding the museum record, it was unearthed from the same Chu tombs of the “Sword of Gou Jian” in 1965 and dated back to the late Spring and Autumn Period (496 BC–464 BC). Obviously, this sword exhibited a similar design philosophy, with the well-recognised bi-metallic bronze sword, i.e., the low-Sn bronze for the mid-ridge to provide good toughness and flexibility and the high-Sn bronze for the blade to provide high strength and hardness. Obviously, the technique for manufacturing this novel kind of composited bronze sword was more advanced than the two-times casting for the regular bi-metallic bronze sword. However, how was this surface treatment made? What were the processing conditions, such as temperature, length of time used, and special additive agent? This work analysed and discussed these issues from a materials science perspective, i.e., element diffusion theory, expecting to reveal a novel approach and machining process used for making bronze swords in ancient China more than 2500 years ago.

## 2. Materials and Experimental Methods

Figure 2 shows a remnant of the bronze sword. It was from the Jingzhou museum of Hubei Province, China. According to the record, it was unearthed from the same Chu tombs of the “Sword of Gou Jian” in 1965; that is to say, it was made during the late Spring and Autumn Period (496 BC–464 BC). However, because it is a broken sword and was not studied previously, the museum could not provide more archeological information. Macroscopically speaking, the late Spring and Autumn Period was an age of high-level bronze technology in the Wu, Yue and Chu states of ancient China, especially regarding swords and other weaponry due to frequent wars.

The metallographic cross-section sample was cut along the vertical direction of the bronze sword fragment and then mechanically polished and etched according to regular standard procedures, i.e., mounting, grounding, polishing and then etching by dipping into alcoholic ferric nitrate solution. The etching solution was made of 2 g ferric nitrate and 50 mL ethanol. The fractographic cross-section sample was directly broken at room temperature or a low temperature, which involved cooling using liquid nitrogen.

The metallographic micrographs were observed by using an optical microscope (Cover-18, Olympus, Tokyo, Japan). Microstructural observations and chemical composition analysis in spot- and line-scan modes were carried out by using a scanning electron microscope (SEM, Sirion, FEI, The Netherlands) equipped with an energy-dispersive X-ray spectrometer (EDS, Genesis 7000, EDAX Inc., Warrendale, PA, USA) operated at 15 kV. The spot-scan mode with five spot measurements was used to quantitatively obtain the average composition values of the sword, while the line-scan mode was used to qualitatively obtain the profiles of the Sn content variation along the cross-section from the sword’s edge to the body.

The microhardness was measured using a Vickers microhardness tester (HXD-1000, Shanghai Taiming Optical Instrument Co., LTD, Shanghai, China) with a load force of 1.961 N (200 g) and a force-holding time of 15 s.

## 3. Experimental Results

Through systematic characterisations involving microstructures, chemical compositions and microhardness, this bronze sword exhibited the following specialties:

(1) From the appearance, there was no corrosion substance and there was a shiny black surface with a very sharp blade after 2500 years. Initially, the sword colour might have been silvery white or silvery yellow, and the current dark gray colour was the result of thin oxidation or dirt covering that accrued during long-term burial.

(2) Scanning electron microscope (SEM) observations showed that there was a homogeneous layer with a special microstructure, which was obviously different from the body of the sword, as shown in Figure 3. It exhibited the following notable features: (i) the thickest layer, about 0.6 mm, was at the blade, and the ridge was in the range of 0.1–0.3 mm; (ii) there was a clear interface between the layer and the substrate, which had a close combination, though the interface exhibited a saw-toothed variation.

(3) Figure 4 and Table 1 give the chemical compositions measured in line-scan and spot-scan modes, respectively. Obviously, the Sn line-scan profiles qualitatively revealed a higher Sn value in the Sn-rich layer than that in the sword body. The spot-scan quantitative measurements indicated that the sword body was a pure Cu-Sn alloy with a lower Sn content, while the surface layer (named as “Sn-rich layer”) contained not only higher Sn content but also impurity elements, such as Fe, Si and P, which are discussed below.

(4) Figure 5 shows the microstructures and morphologies of the sword cross-section surface. The Sn-rich layer was composed of a large amount of α + δ eutectoid and a few fine and dense α-Cu solid solutions in which few transcrystalline microcracks could be observed. However, the sword body consisted of homogeneous tempered microstructures as an α-Cu phase (bright), α + δ eutectoids (grey) and retained β-Cu phase (acicular structure), as shown in Figure 6d, while the cast microstructures, such as dendrites and columnar grains, were not observed.

(5) Figure 6 shows the fractographies of the sword cross-section surface. The sword body showed a good toughness with many tearing edges, while the Sn-rich layer exhibited a typical brittle fracture. In addition, the features of cleavage fracture, such as transgranular or intercrystallite structures, and even the “rock sugar fast” morphology of intergranular fracture, could be observed through SEM at high magnification, as shown in Figure 6d. These results revealed that the Sn-rich layer was of higher hardness and brittleness than the body, which, in fact, also corresponded to their microstructures, as shown in Figure 5: (i) For the sword body, the α-Cu solid solution with a face-centered cubic (FCC) crystal contributed to the ductile fracture and formed the tearing edges, while the Sn-rich α + δ eutectoid produced the brittle cleavage fracture. (ii) For the Sn-rich layer, the fine and compact Sn-rich α + δ eutectoid would inevitably produce the cleavage brittleness fracture because of the existence of the hard but brittle δ-Cu phase (Cu_31_Sn_8_). In general, the δ-Cu phase not only contains about 32.6 wt.% Sn but is also a complex cubic structure. These fracture morphologies also provided evidence that the Sn-rich layer was not a loose corrosive layer.

(6) Figure 7 illustrates the Vickers microhardness (HV) profile of the sword cross-section surface. The highest hardness of the Sn-rich layer reached over 650 HV, while the sword body was only around 300 HV. Obviously, this also demonstrated that the Sn-rich layer was not a loose corrosive layer.

From the above experimental results, it could be confirmed that this bronze sword had undergone surface treatment, i.e., a surface modification (thermal hardening), and not a surfacing coating. It also demonstrated that this bronze sword was displayed good performance with a combination of toughness and strength as a well-recognised “double-colour” or bi-metallic composite bronze sword. 

Comparatively, the manufacturing technique involving surface modification to form a Sn-rich layer in the sword surface provides more advantages: (1) The Sn-rich surface treatment not only possesses much high hardness for cutting, but it also greatly increases the corrosion resistance. (2) The manufacturing technique does not affect the sword appearance, when compared with the two-times casting “double-colour” sword. (3) Finally, the silvery-white or silvery-yellow surface also makes the sword more beautiful and dignified.

## 4. Simulation Calculations of Sn Element Diffusion in Bronze

As discussed in our previous work [7], a high-Sn region on a bronze surface could arise from a method, i.e., a kind of deliberate tinning of bronze, such as simple dip or wipe tinning. Therefore, for this sword, a possible process would include the steps of dip tinning or Sn amalgamation on the sword surface at high temperature after casting, keeping it for a while to form a Sn-rich layer, and finally grinding and finishing the sword.

According to the basic principle (theory) of materials science, the essence of this process is the diffusion of Sn atoms into the Cu (or Cu-Sn alloy) substrate at a high temperature. Experiments confirmed that when the temperatures were above 450 °C, Sn atoms obtain enough energy to diffuse into the Cu substrate, and the diffusion length is determined by temperature and time in which the (α + δ) eutectoid forms [8]. In other words, due to the difference in concentration, Sn atoms diffuse into the bronze substrate at a high temperature and form a layer of dense and high-hardness Cu-Sn phases in the surface, namely, the “Sn-rich layer”. 

Therefore, we explored the possibility and feasibility of the surface treatment with the temperatures and times that are used for forming a Sn-rich layer in a range of 0.1–0.6 mm on the bronze surface via theoretical calculations according to the basic law of diffusion, namely, Fick’s law.

### 4.1. The Basic Law of Element Diffusion

(1) Fick’s first law: A quantitative formula for describing the migration of matter from a region of high concentration to a region of low concentration:$$J=-D\frac{\partial c}{\partial x}$$

In the one-dimensional case, *J* is the diffusion flux. It is the number of atoms passing through the unit plane perpendicular to the *x*-axis in a unit of time. *∂c/∂x* is the concentration gradient along the *x*-axis for the same time. *D* is the diffusion coefficient, which represents the flux under a unit gradient. This equation is applicable to not only any position of the diffusion system but also any time in the diffusion process. It is suitable for both steady-state and unsteady-state diffusion systems.

(2) Fick’s second law: When the diffusion is in an unsteady state, i.e., $\frac{\partial c}{\partial t}\;\neq 0$, the change rate of the concentration with time at a certain point during diffusion is proportional to the second derivative of the concentration distribution curve at that point. In the case of one-dimensional diffusion, the diffusion process satisfies the following equation:$$\frac{\partial c}{\partial t}=\frac{\partial }{\partial x}(D\frac{\partial c}{\partial x})$$
where its general solution is
$$C=a\int_{0}^{\beta }\exp(-\beta^{2})d\beta +b$$
in which $\beta =x/(2\sqrt{Dt})$.

When diffusing within a semi-infinite object, the values of *a* and *b* can be determined from the initial concentration and boundary concentration. Then, regarding the concentration distribution together with the diffusion distance after diffusion, the diffusion time (t) can be calculated.

### 4.2. Calculation of Sn Diffusion in Bronze

In general, as a kind of surface engineering technology, the thermochemical process produces a high-hardness surface layer in a steel substrate via non-metal (C, N, C + N, P, Si, etc.) and metal (Cr, Al, V, Ti, etc.) diffusions at high temperature. In order to further estimate the diffusion thickness, Fick’s laws are used for the theoretical calculation, and for simplicity, a simple model was established, i.e., one kind of atom diffused into the pure metal substrate in one direction. In fact, the grain boundaries and phase boundaries in the diffusion layer can accelerate the diffusion rate, which is generally included in the diffusion coefficient (D). 

Regarding the “dip or wipe tinning” process, the bronze sword was dipped in a molten Sn pool or coated with a Sn powder. Then, Sn atoms started to diffuse into the bronze sword substrate during reheating for a period of time at high temperatures, and lastly, a dense Sn-rich layer was obtained on the surface. 

As discussed in Meeks’ work [8], the “dip or wipe tinning” process produced a complicated phase mixture, such as α-Cu phase, η-Cu phase, ε-Cu phase and δ-Cu phase, during the growth of the Sn-rich layer. Some of these phases belong to a kind of metastable structure and would transform into stable phases during high-temperature treatment. Therefore, for the thickness estimation of the Sn-rich layer, the above simple assumption was also applied, which is commonly used in the thermochemical process; that is to say, the influence of the intermetallic compounds on the Sn-rich layer formation was included in the values of the Sn diffusion coefficient (*D*).

For the present work, the assumption of the calculation model was as follows: (1) Sn atoms diffused in a pure Cu substrate at a certain temperature; (2) the Cu substrate was one-dimensional and had a semi-infinite length; (3) the time (t) represented the length of the period for forming a 0.1–0.6 mm diffusion layer (Sn-rich layer); and (4) only the concentration gradient was considered for Sn atoms migration, i.e., downhill diffusion.

Therefore, Fick’s second law could be solved from the Boltzmann transform:$$\frac{\partial c}{\partial t}=\frac{\partial }{\partial x}(D\frac{\partial c}{\partial x})$$
which has the general solution:$$C=a\int_{0}^{\beta }\exp(-\beta^{2})d\beta +b$$
in which *a* and *b* are the constants to be determined, and $\beta =x/(2\sqrt{Dt)}$.

Regarding the diffusion of a semi-infinite object, the surface element concentration remained constant. The length of the object was greater than $4\sqrt{Dt}$, which was similar to that of the process, such as metal carburising or nitriding treatment; that is, Sn first reached a limiting concentration at the surface when other elements diffused into the metal and then remained constant. Meanwhile, Sn kept diffusing continuously in the metal substrate; that is, the initial conditions: *t* = 0, *x* > 0 and *c* = 0; and the boundary conditions: *t* ≥ 0, *x* = ∞, *c* = 0; *x* = 0, *c* = *C*_0_.

After substituting the boundary conditions and initial conditions into the general solution, the relationship of the thickness, concentration and time was obtained:$$C=C_{0}(1-erf(\beta ))$$
in which
$$\beta =x/(2\sqrt{Dt})\;erf(\beta )=\frac{2}{\sqrt{\pi }}\int_{0}^{\beta }\exp(-\beta^{2})d\beta$$

This was the limiting concentration of the diffused elements at the surface of the metal substrate.

Because the Sn-rich layer was composed of a large number of (*α* + *δ*) eutectoids and few α-Cu phases, an approximate consideration of the Sn-limiting concentration on the surface of Cu substrate was the value in the *δ*-Cu phase, i.e., *C*_0_ = 32.6 wt.%.

The experimental results revealed that the thickest Sn-rich layer on the blade was about 0.6 mm, while the thickness of the ridge was in a range of 0.1–0.3 mm, and the average Sn content in the sword body was 18.57 wt.%; that is to say, when the diffusion distances (*x*’s) were 0.1 mm, 0.2 mm, 0.3 mm and 0.6 mm, the Sn concentration (*C*) was 18.57 wt.%. In other words, *erf*(*β*) was obtained and the *β* was 0.4 from the tabulation of error function values [9] due to the known values of *C*_0_ and *C* in the equation. Consequently, the diffusion time (*t*) behaved according to a function of the diffusion coefficient (*D*) and diffusion distance (*x*) as follows: $$t=\frac{x^{2}}{4{D\beta }^{2}}$$

However, in the present case, this diffusion is very difficult because the radius of Sn atoms (140 pm) is greater than that of Cu atoms (128 pm), and the diffusion of Sn in the Cu substrate demonstrates the substitutional mechanism; that is to say, the diffusion needs a high temperature and a long time. In addition, the diffusion coefficient (*D*) is different at different temperatures, and the *D* values increase with the temperatures. Referring to the book “Smithells Metal Reference Book” (7th edition) [10], the *D* values at several different temperatures are listed in Table 2. For example, regarding the data temperature at 700 °C, *D* = 4.3 × 10^−9^ cm^2^/s and *x* = 0.2 mm, the diffusion time (*t*) was calculated to be about 41 h using the diffusion time formula.

Similarly, Table 2 gives the calculation results of Sn diffusion in the Cu substrate. It revealed the relationships between temperatures, times (*t*’s) and diffusion distances (*x*’s). Figure 8 illustrates the corresponding fitting curves of the required diffusion times of the Sn-rich layer with different thicknesses as a function of temperature. Obviously, with an increase in temperature, the diffusion time (*t*) reduced rapidly for the same layer thickness, i.e., the higher the temperature, the faster the diffusion rate.

Considering the possible formation in entombment during thousands of years, Figure 9 illustrates the curing that occurred in such conditions, such as 2000 years, room temperature, *D* = 10 × 10^−17^ cm^2^/s [11] and the same boundary concentration of 32.6%. It showed that the diffusion distance was less than 0.05 mm, which was very small and could be ignored.

## 5. Discussion

Regarding Table 2, as an example, the time for forming a 0.2 mm thickness of the Sn-rich layer at 600 °C required 11 days. Figure 10 illustrates a corresponding profile of the Sn concentration variation with distance from the diffusion formula. They were basically consistent when compared to the experimental profile from the EDS line-scan, as shown in Figure 11. In other words, the presented theoretical calculation was reasonable.

In addition, there was a “platform” with a constant Sn content in the Sn-rich layer from the line-scan. The cause of such a phenomenon could have been as follows. The surface was rapidly saturated with Sn when the diffusion began and reached the value as in δ-Cu phase, i.e., forming δ-Cu phase on the surface. Furthermore, the δ-Cu phase kept growing and formed an initial Sn-rich layer with a full δ-Cu phase adjacent to the surface as the Sn diffusion continued, which resulted in the maximum Sn content, i.e., the “platform”. Afterwards, the Sn concentration gradually decreased.

Another phenomenon was that there was obviously a downward step between the Sn-rich layer and the Cu substrate at the interface in the Sn line-scan for all areas. According to the phase diagram of a Cu-Sn binary alloy, the diffusion process not only led to the formation of an α-Cu solid solution and compositional changes but also resulted in other phase transformations, such as δ-Cu and β-Cu. Therefore, this content drop at the interface because of the formation of different phases due to high Sn content in the Sn-rich layer.

To summaries, from the above theoretical calculation, it was demonstrated that to have a Sn-rich layer in a range of 0.1 mm–0.6 mm, the time for the surface treatment should be as short as few hours or as long as a few days at different temperatures. The thickest layer (0.6 mm) in the blade (the edge) might have been due to the “tip or edge effect” during dipping at a high temperature. Let us imagine during the Bronze Age that it was possible and worthwhile for an ancient technician to spend time manufacturing a high-quality bronze sword due to the slow rhythm of life. In fact, we also found a few of this kind of bronze sword in our previous work [12]. Sun et. al. [13] reported and analysed 47 tinning bronze weapons, which were unearthed from broad areas, such as Guyuan in Ningxia province, Chengdu and Mianzhu in Sichuan province and Yunnan province in China, dated to the fifth-second century B.C., i.e., from the Spring and Autumn and the Warring States Periods to the early Western Han dynasty. However, the tinning layers were thin, only within a range of 20–40 μm mostly, which might have been for the purpose of decoration and corrosion resistance.

The ancient Chinese technicians might have added catalysts or active agents, or energising reagents for accelerating the Sn diffusion process when dip or wipe tinning, as we do today in the industry; that is to say, the actual making time was much shorter than this calculated data. From Table 1, it was interesting to note that in addition to the main alloys Cu and Sn, there were more or fewer impurity elements, including Fe, Si and P, in the Sn-rich layer. Regarding their contents, these impurities did not seem to come from the surrounding environment, such as soil or water, and might be the residues from the surface treatment. Were they from a special catalyst added in the molten Sn during “dip or wipe tinning”? Possibly. As we know, adding a catalyst is also a common method in modern surface treatment. In addition, some studies proposed that elemental Fe in the Sn-rich layer was induced by using iron tools during the operation of “dip or wipe tinning”, which was very common in the Spring and Autumn and the Warring States Periods [14,15].

It has been recognised that the high-Sn region or layer on the ancient bronze surface due to the following ways [8]: (1) tinning the surface by using processes including simple dip or wipe and mercury amalgam, (2) “tin sweat” produced during solidification and (3) selective corrosion. For the present bronze sword, the dense and compacted fracture surface, as shown in Figure 6c,d, and very high microhardness, as shown in Figure 7, confirmed that the layer was not a loose corrosive substance. Therefore, we proposed that this Sn-rich layer was manufactured via a process of simple dip or a wipe tinning, and its relatively large thickness made it a novel kind of ancient composite bronze sword, which technically possessed more advantages than the common two-times casted bi-metallic composite bronze sword.

## 6. Conclusions

As a widely used surface treatment “dip or wipe tinning” in the Bronze Age, this work demonstrated experimentally and theoretically that it could also be applied to make composite bronze swords besides the general purposes of decoration and anti-corrosion. Compared with the “double-colour” or bi-metallic composite bronze sword made by using two-times coating, this kind of composite sword was of higher technicality and provided more technical support and flexibility. It not only had a comprehensive performance regarding toughness and strength without changing the sword’s appearance but also made the sword more beautiful and dignified. Such a composite sword overturns our perception of the ancient bronze manufacturing technique. In addition, it also inspires us to study the ancient bronzes with professional knowledge of material science in the future.

## Figures and Tables

**Figure 1 materials-15-02491-f001:**
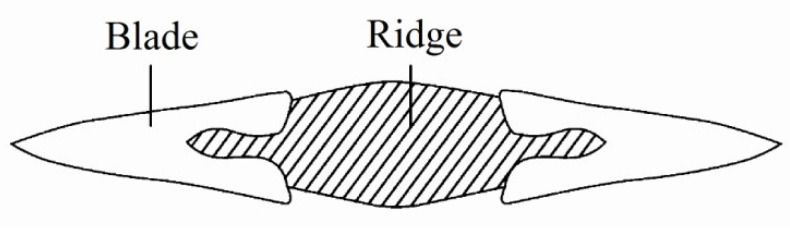
Sketch diagram of the two-times casting for making the bi-metallic bronze sword.

**Figure 2 materials-15-02491-f002:**
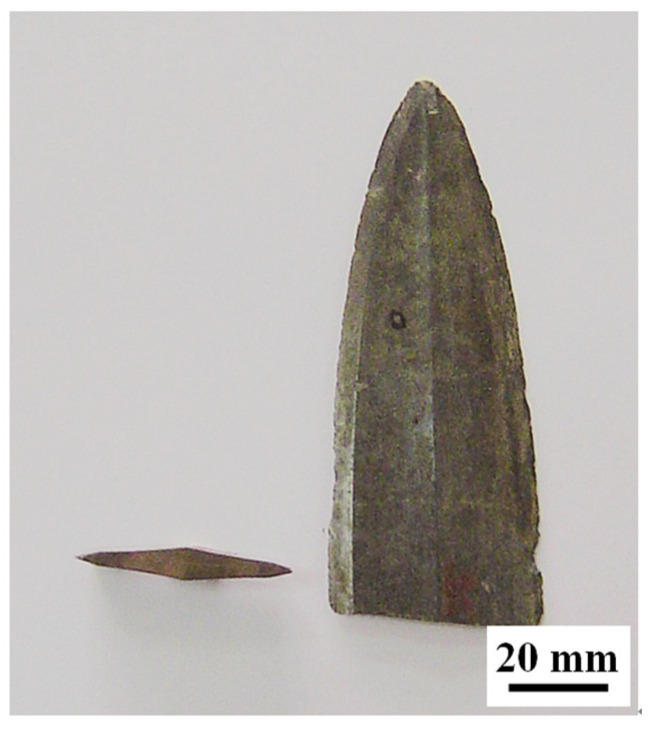
Images of the remnant of the bronze sword.

**Figure 3 materials-15-02491-f003:**
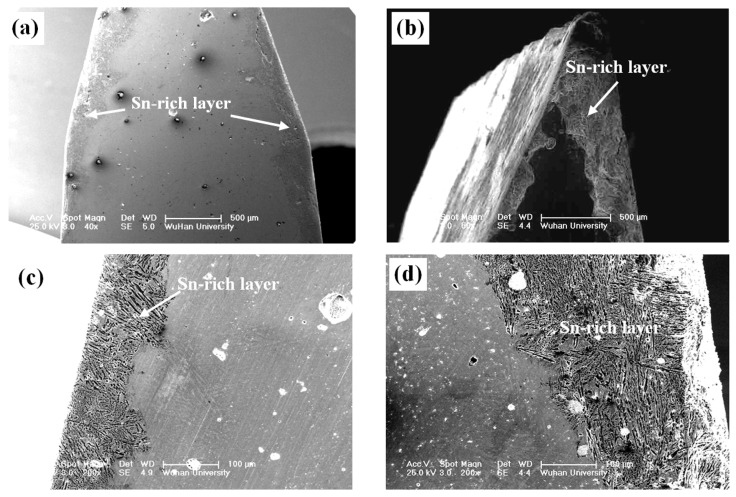
Cross-section SEM morphologies of the sword: (**a**,**b**) low magnification; (**c**,**d**) high magnification.

**Figure 4 materials-15-02491-f004:**
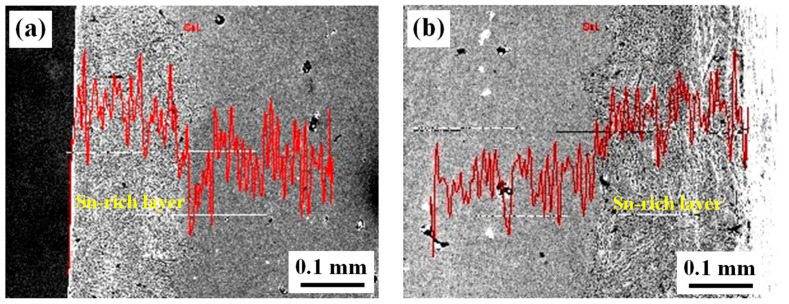
Cross-section EDS line-scan profiles of the sword: (**a**) left side; (**b**) right side.

**Figure 5 materials-15-02491-f005:**
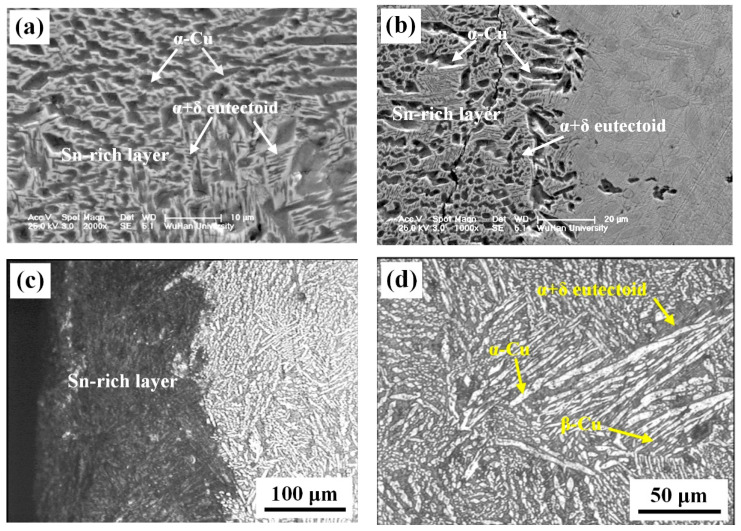
Cross-section microstructures of the sword: (**a**) SEM morphology of the Sn-rich layer; (**b**) SEM morphology of the interface between the Sn-rich layer and the body; (**c**) optical image of the interface between the Sn-rich layer and the body; (**d**) optical image of the body.

**Figure 6 materials-15-02491-f006:**
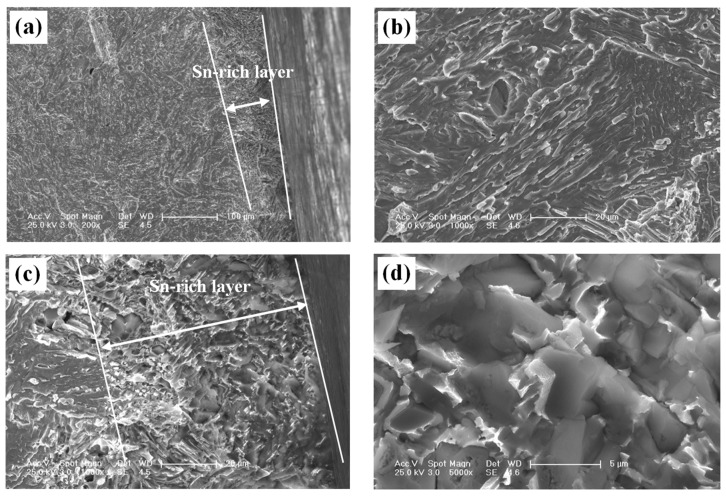
Cross-section fracture surface SEM morphologies of the sword: (**a**) low magnification; (**b**) high magnification of the body; (**c**) portion of the Sn-rich layer; (**d**) high magnification of the Sn-rich layer.

**Figure 7 materials-15-02491-f007:**
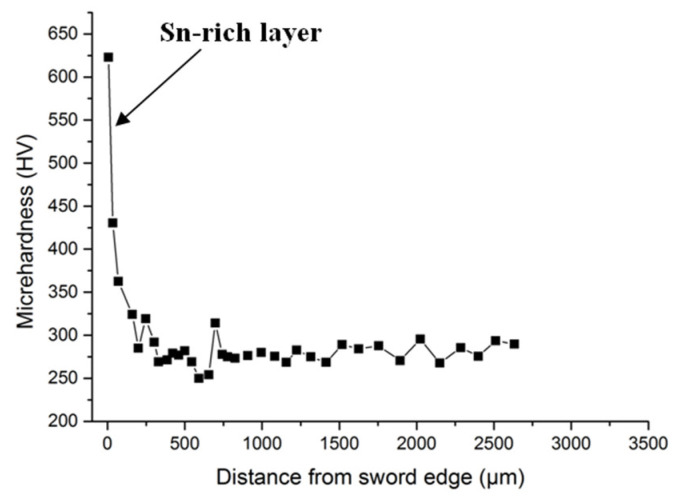
Vickers microhardness profile of the sword cross-section surface.

**Figure 8 materials-15-02491-f008:**
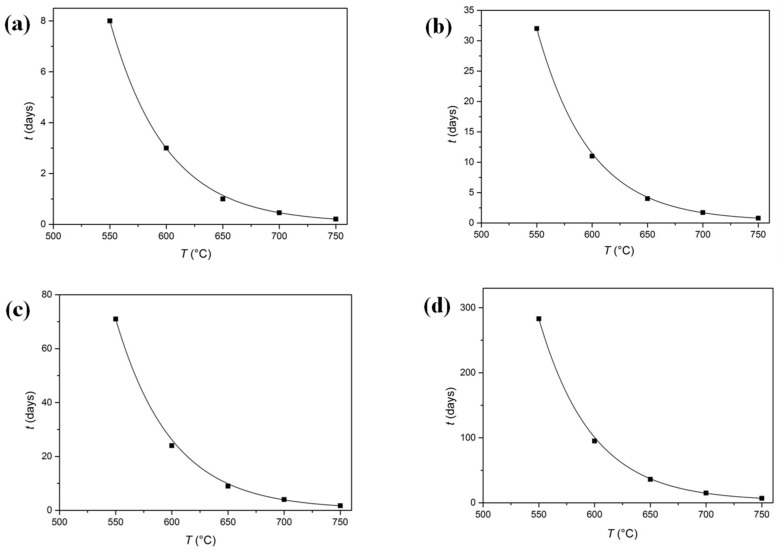
Fitting curves of the required diffusion time of the Sn-rich layer with different thicknesses as a function of temperature: (**a**) 0.1 mm; (**b**) 0.2 mm; (**c**) 0.3 mm; (**d**) 0.6 mm.

**Figure 9 materials-15-02491-f009:**
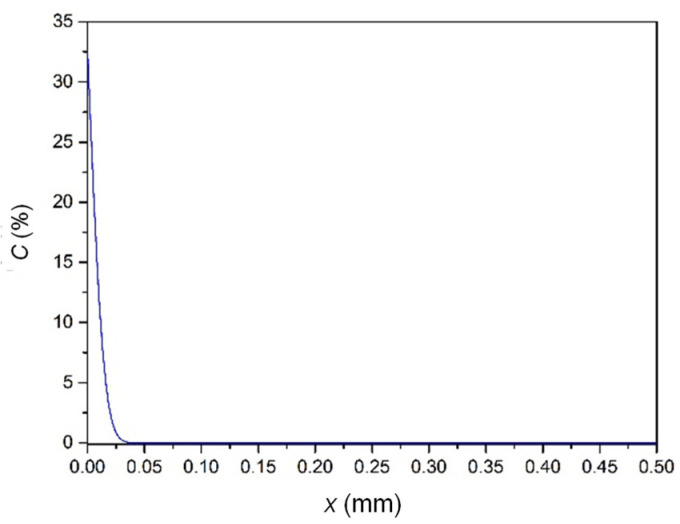
Fitting curves at room temperature for 2000 years.

**Figure 10 materials-15-02491-f010:**
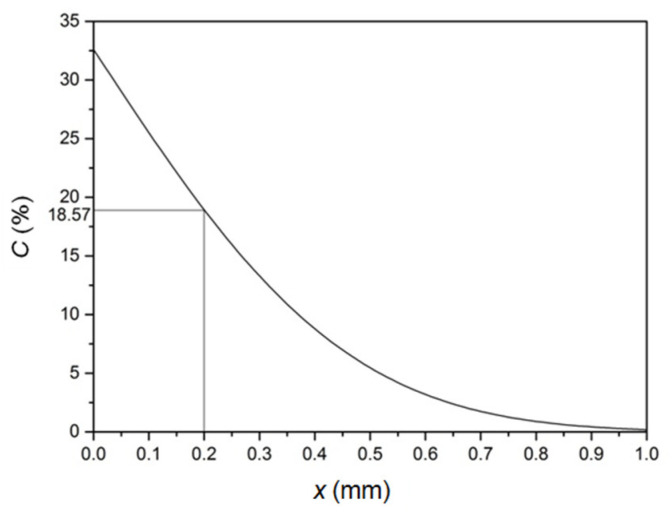
Calculated concentration–distance (*C*–*x*) profile at 600 °C in the case of a 0.2 mm thickness of the Sn-rich layer.

**Figure 11 materials-15-02491-f011:**
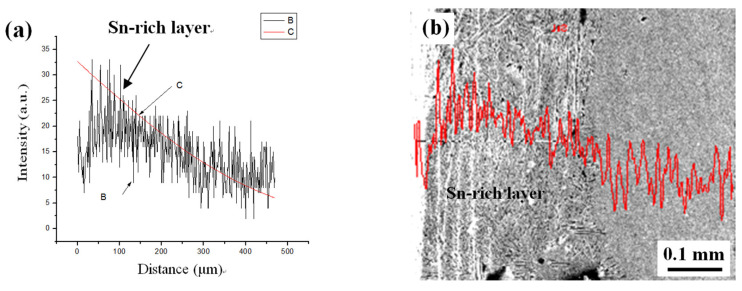
Comparison between the computational result (**a**) and the experimental EDS line-scan profile (**b**) (B—experimental profile, C—theoretical profile).

**Table 1 materials-15-02491-t001:** Chemical compositions of the sword (wt.%).

Element	Cu	Sn	Fe	Si	P
Sword body	81.43	18.57	—	—	—
Sn-Rich layer	53.42	38.51	6.06	1.27	0.75

**Table 2 materials-15-02491-t002:** Calculation results of Sn diffusion in the Cu substrate.

Temperature (°C)	D (cm^2^/s)	Time (*x* = 0.1 mm)	Time (*x* = 0.2 mm)	Time (*x* = 0.3 mm)	Time (*x* = 0.6 mm)
550	2.3 × 10^−10^	8 d	32 d	71 d	283 d
600	6.9 × 10^−10^	3 d	11 d	24 d	95 d
650	1.8 × 10^−9^	1 d	4 d	9 d	36 d
700	4.3 × 10^−9^	11 h	41 h	4 d	15 d
750	9.5 × 10^−9^	5 h	19 h	41 h	7 d

## Data Availability

Data is contained within the article.

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
