# Peer review of "A Special Ancient Bronze Sword and Its Possible Manufacturing Technique from Materials Science Analysis"

_materials, 2022, doi:10.3390/ma15072491_

Round 1

Reviewer 1 Report

APPRECIATION OF THE SCIENTIFIC INTEREST AND NOVELTY

  • The authors analyze the Sn-rich surface of a bronze sword
  • Based on experimental measurements and 1-d diffusion model a model of the fabrication process is proposed

COMMENTS TO AUTHORS

This manuscript provides historical context and description of bronze swords in historical periods in China.  The content is primarily descriptive and contains very limited quantitative results.  As such, it seems more appropriate for a journal focused on archeology or history rather than this forum.

The data presented consists primarily of scanning electron microscopy (SEM) images which provide only qualitative information about the structure of the material being studied.  The authors state the sword contains a number of different metallic phases intermixed in the various regions but there are no measurements to verify these assertions and these regions are not identified in the images provided.  A change in content between the high- and low-Sn regions is demonstrated via energy dispersive spectroscopy, but no quantitative scale is provided.

Quantitative data is provided for the hardness of the high- and low-Sn regions.  These results are used to explain the motivation for this sword design.

A 1-d diffusion model is provided to estimate the time required to achieve the thickness of the high-Sn coating layer observed.  The resulting values, ranging from 5 hours for the thinnest regions to 7 days for the thickest regions seem impractical.  In addition, there is no discussion regarding why or how such a large difference in thickness is observed.  Furthermore, the authors discount their own model by suggesting catalysts may have been in use.

The axes on Fig. 12a are not labeled. 

Author Response

COMMENTS TO AUTHORS

1. This manuscript provides historical context and description of bronze swords in historical periods in China.  The content is primarily descriptive and contains very limited quantitative results.  As such, it seems more appropriate for a journal focused on archeology or history rather than this forum.

Reply: Thanks for the comment! In recent several tens of years, many kinds of fields in science and engineering have been getting involve in the archeology or history, such as chemistry, physics, materials science, plant, anthropology, geophysics, molecular biology and environmentology, etc. and promoted the development of archaeology. As far as I know, the research achievements are published in variant journals, not only archeology or history journals, but also specialized professional journals. In the past 17 years, my group also published several tens of paper in journals on both archeology, physics as well as materials science.

In terms of professional field, ancient bronze or Cu-Sn alloy belongs to materials science and technology. Actually, this paper contains few archeology information, and most results and discussion are related to materials physics. Our purpose for publishing this work in the Materials of this forum is to get the value feedback from the materials experts, and also expect more materials scientists joint the research area of ancient bronzes.

2. The data presented consists primarily of scanning electron microscopy (SEM) images which provide only qualitative information about the structure of the material being studied.  The authors state the sword contains a number of different metallic phases intermixed in the various regions but there are no measurements to verify these assertions and these regions are not identified in the images provided.  A change in content between the high- and low-Sn regions is demonstrated via energy dispersive spectroscopy, but no quantitative scale is provided.

Reply: Thanks for the comment! In general, ancient bronze is Cu-Sn alloy, which contains few alloys and has simple phase transformations. Many early researches have fully and quantitatively studied the microstructural characters of different metallic phases in different conditions, such as temperature, cooling rate and chemical compositions. These results have been widely accepted and recognized by researchers, and, therefore, the subsequent work generally need not to repeat the experiments. In order to easily identify the phases, we add the indications in the Figure 5 in the revised manuscript.

3. A change in content between the high- and low-Sn regions is demonstrated via energy dispersive spectroscopy, but no quantitative scale is provided.

Reply: In principle, EDS line-scan mode can only give a qualitative result of the composition variation, which generally gives a relative value difference. The line variations actually are the intensity counting rates from the measured region. In this work, the most important is to know the tendency of the Sn element distribution, which provide the basic data for supporting the subsequent theoretical diffusion calculation. This experimental method is widely used in the thermochemical process of surface modifications, such as non-metal (C, N, C+N, P, Si, etc.) and metal (Cr, Al, V, Ti, etc.) diffusion at high temperature in the steel substrate for getting a high-hardness surface layer.

4. Quantitative data is provided for the hardness of the high- and low-Sn regions. These results are used to explain the motivation for this sword design.

Reply: Thanks! These results are very important, because it excludes the Sn-rich layer a loose corrosion layer, which generally results in a collapse during hardness testing. That is to say, the high-Sn region was intended to be produced for getting extra-high hardness. This process is of the similar principal of the modern thermochemical process of surface modification.

5. A 1-d diffusion model is provided to estimate the time required to achieve the thickness of the high-Sn coating layer observed.  The resulting values, ranging from 5 hours for the thinnest regions to 7 days for the thickest regions seem impractical.  In addition, there is no discussion regarding why or how such a large difference in thickness is observed.  Furthermore, the authors discount their own model by suggesting catalysts may have been in use.

Reply: Thanks for the very good comments! In this work, we want to propose a possibility experimentally and theoretically that the ancient technicians had knew and mastered the Sn surface treatment technique. In fact, the present data errors may come from several factors, for example, simple 1-d diffusion model, large Sn diffusion coefficient in different temperature, and really manufacturing conditions at that time, etc.

The innovation point of this paper is the first time to use the diffusion theory to study the ancient bronzes. It seems reasonable.

Currently, we are still working on this area, especially, lots of work on simulation of the Sn diffusion in Cu-Sn ally are carrying on. We expect to have the valuable results in our next paper soon.

6. The axes on Fig. 12a are not labeled. 

Reply: Thanks! The labels have been added in the figure in the revised manuscript.

Reviewer 2 Report

Dear Authors,

I found you paper very interesting. However some changes are suggested in the pdf.

Very important: a “Methods/Experimental” section must be added in the manuscript. This section must include the characteristics of all equipment used and a clear description of all methodologies employed.

Best wishes

Author Response

Thank for the very helpful and professional comments! It could be seen that all the comments have been accepted and the relative contents have been corrected in the revised manuscript. Following give the main changes as examples.

Comment 1: Keep just one scale for each figure.

Reply: The added scales have been deleted from the related figures.

Comment 2: This section should be named "Results" and should be preceded by a "Methodology" section. Consequently, the structure should be modified as followings: 1. Introduction; 2. Methods; 3. Results ....

Reply: The revised manuscript has been changed to the structure as the review’s suggestion. And more information about the experimental conditions of analysis also have been provided in the revised manuscript.

Reviewer 3 Report

In this paper, a special bronze sword was found, which had an Sn-rich layer in the bronze substrate, with the Sn-rich layer exhibiting much higher hardness than the sword body. The subject is particularly interesting for the special issue but limited to general Materials Science, in special physical metallurgy. I do not see knowledge addition to physical metallurgy but in metallurgy’s history. The methodology is usual for this kind of study and material. The results are very interesting, sound, and clear for the paper’s purpose. The discussion of the results is coherent and based on the earlier findings in the bronze metal. The conclusions are based on the obtained results and presented discussion. From the historical point of view, the paper is adequate, but scientifically, little information will be aggregated to the physical metallurgy field.

Reviewer 4 Report

The article is very interesting from the point of view of the knowledge of products from our past and the method of their production. From the point of view of the origin of the sword - everything is explained well and you can form an opinion about the history of this element. The research is also carried out comprehensively and properly presents the current properties of the sword.
As for the comments, I miss such a detailed explanation of the method of production of this sword, so as to obtain just such zones in terms of chemical composition, one rich in Sn, which has completely different properties than the rest of the material. It turns out that these properties are much better from the point of view of the requirements for the sword. It might be tempting to analyze the manufacturing technology in more detail than what is in the article,
Congratulations on your interesting work, he is a bit interested in history himself, especially in the properties of materials from the time when they were just about to be made - and they had to be given some properties after all.

Round 2

Reviewer 1 Report

The manuscript has been substantially improved, however, the primary limitation previously noted still remain.  Namely, the manuscript is predominantly descriptive and does not represent State-of-the-Art Materials Science.

In their response to the initial reviewer comments, the authors note that EDS provides qualitative results not quantitative values.  However, in the manuscript, the authors present quantitative values of composition (to .01% significance!) based on EDS line and spot scans.  These data are subsequently compared to the results of a 1-d diffusion model to determine the duration of the proposed surface treatment process.  Much of the analysis is based on conjecture with little or no supporting evidence.

The manuscript still contains typographical errors and poor language usage.

Author Response

The manuscript has been substantially improved, however, the primary limitation previously noted still remain.  Namely, the manuscript is predominantly descriptive and does not represent State-of-the-Art Materials Science.

Reply: In fact, the special Issue “State-of-the-Art Materials Science in China” is expecting to include broad research areas in materials in China. In recent years, conservation and research of the Chinese cultural relics have been the national strategic goal. As a kind of materials, the ancient bronze’s conservation and its technological development have received high attentions not only by archaeologists but also by materials scientists. Its new achievements also represent a research direction of materials development in China.

In my opinion, this paper is of two innovations, i.e., 1) It is firstly found a new kind of ancient composite bronze sword via surface treatment, which is different from the well-known two-times casting for the bi-metallic bronze sword and also exhibits the similar property; 2) The possible manufacturing process is theoretically analyzed according to the materials physics. This work actually takes the ancient bronze research one step further. 

In their response to the initial reviewer comments, the authors note that EDS provides qualitative results not quantitative values.  However, in the manuscript, the authors present quantitative values of composition (to .01% significance!) based on EDS line and spot scans.  These data are subsequently compared to the results of a 1-d diffusion model to determine the duration of the proposed surface treatment process.  Much of the analysis is based on conjecture with little or no supporting evidence.

Reply: Regarding the query, I think the reviewer has a misunderstanding. I would like to have more explanations.

A commercial SEM+EDS system generally provides two modes for measuring the compositions, i.e., spot-scan mode and line-scan mode.

1)The spot-scan mode can give the quantitative values in a spot, which actually is a very small area due to X-ray excitation in depth and also influenced by other factors (ZAF effects). During the measurement, in order to have the data as accurate as possible, it generally takes several spots in different places to get the average quantitative values of materials. The data in Table 1 was measured by using the spot-scan mode.

2) The line-scan mode can only give a qualitative value along a selected line, because the line height (y-axis) is the intensity counting rates of X-ray photons from the measured region, not the contents, such as wt.% or at.%. The line fluctuation relates to the phases, grain boundary, cavity and impurity, etc. One line represents one element (Figure 4), and several elements can also be measured simultaneously.

Practically, due to its fast and direct features, the line-scan mode is commonly used for getting the variation tendency of the compositions, such as segregation and dissimilar materials connections (welding and composite, etc.). In the actual analysis, if we know quantitatively the initial value or last value (Table 1), the composition variation along the selected line can be qualitatively estimated accordingly. This is why we use these experimental results to compare the results of a 1-d diffusion model to determine the duration of the proposed surface treatment process, as shown in Figure 11. 

Above messages are commonly used in materials analysis. In order to let the readers, who do not work in materials science, to clearly understand this paper’s results, I added more messages in the revised manuscript,such as:

(1) "The spot-scan mode with five spot measurements was used to quantitatively obtain the average composition values of the sword, while the line-scan mode was used to qualitatively get the profiles of the Sn content variation along the cross-section from the sword's edge to the body." (in section "Materials and experimental methods");

(2) "Obviously, the Sn line-scan profiles qualitatively revealed a higher Sn value in the Sn-rich layer than that in the sword body. The spot-scan quantitative measurements indicated that the sword body was pure Cu-Sn alloy with relative lower Sn content, while the surface layer (named as "Sn-rich layer") contained not only higher Sn content, but also impurity elements, such as Fe, Si and P, which will be discussed below". (in section "Experimental results" 3))

Also, the basic principle of EDS analysis could be find in many references and EDS instruction manuals. Here, I list two books for review’s and editor’s reference.

[1] V. D. Scott and G. Love (editors): Quantitative Electron-Probe Microanalysis, published in 1983 by Ellis Horwood Limited.

[2] Jing Zhu, Hengqiang Ye, Renhui Wang, Shulin Wen and Zhenchuan Kang: High Spatial Resolution Analytical Microscopy, Science Press, 1987, in Chinese.

 The manuscript still contains typographical errors and poor language usage.

Reply: Thanks for the comment. The manuscript is carefully checked again in language writing. It is expected to meet the publishing level.